# Tunable Split-Disk Metamaterial Absorber for Sensing Application

**DOI:** 10.3390/nano11030598

**Published:** 2021-02-27

**Authors:** Yusheng Zhang, Peng Lin, Yu-Sheng Lin

**Affiliations:** School of Electronics and Information Technology, Sun Yat-Sen University, Guangzhou 510006, China; zhangysh23@mail2.sysu.edu.cn (Y.Z.); linp25@mail2.sysu.edu.cn (P.L.)

**Keywords:** tunable metamaterial, metal-insulator-metal absorber, plasmonic sensor

## Abstract

We present four designs of tunable split-disk metamaterial (SDM) absorbers. They consist of a bottom gold (Au) mirror layer anchored on Si substrate and a suspended-top SDM nanostructure with one, two, three, and four splits named SDM-1, SDM-2, SDM-3, and SDM-4, respectively. By tailoring the geometrical configurations, the four SDMs exhibit different tunable absorption resonances spanning from 1.5 µm to 5.0 µm wavelength range. The resonances of absorption spectra can be tuned in the range of 320 nm, and the absorption intensities become lower by increasing the gaps of the air insulator layer. To increase the sensitivity of the proposed devices, SDMs exhibit high sensitivities of 3312 nm/RIU (refractive index unit, RIU), 3362 nm/RIU, 3342 nm/RIU, and 3567 nm/RIU for SDM-1, SDM-2, SDM-3, and SDM-4, respectively. The highest correlation coefficient is 0.99999. This study paves the way to the possibility of optical gas sensors and biosensors with high sensitivity.

## 1. Introduction

Recently, gas sensing technology reveals good prospects with the aggravation of environmental pollution. There are mainly two approaches, including electrochemical sensing [1] and nondispersive infrared (NDIR) sensing [2]. The electrochemical sensor has been widely employed in commercial products. However, it is limited by large power consumption, poor selectivity, and hysteresis [3]. Rather than the electrochemical sensor, the infrared (IR) sensor is used to detect gas molecules owing to the specific absorption spectrum of gas molecules in the near to middle IR wavelength. It gradually becomes a mainstream of research for its high selectivity and lower hysteresis, but it still suffers from poor absorption characteristics and selectivity in the mixed environment where various gases and water vapor coexist. In view of this point, some literature has reported using metamaterial absorber with high selectivity to enhance the absorption characteristics in the mid-IR wavelength range [4,5,6,7,8]. Metamaterials, composed of periodic or random structures with subwavelength feature size, exhibit unique electromagnetic and optical characteristics, such as negative refraction index [9], perfect absorption [10,11,12], and so on. Owing to their extraordinary electromagnetic properties, metamaterials are widely explored for optical filters, waveguides, polarizers, resonators, and absorbers [13,14,15,16,17,18] spanned from microwave, terahertz (THz), and IR to visible spectra ranges [10,19,20,21,22].

The various surrounding gas molecules including water vapor have their own absorption spectra, which will make the electromagnetic resonance of the IR metamaterial absorbers shift or decrease the resonance intensity. However, it is difficult to meet the requirements of high selectivity owing to the fact that the electromagnetic characteristics of metamaterial absorbers are fixed once fabricated on rigid substrates. Therefore, an actively tunable metamaterial in the mid-IR wavelength range becomes a feasible approach. The controllable methods of tunable metamaterials include thermal annealing [23,24,25,26,27], phase change materials [28], laser pumping [29,30,31], liquid crystal [32,33], flexible materials [34], and micro-electro-mechanical systems (MEMS) technique [35,36,37,38,39,40]. In these tuning approaches, MEMS-based metamaterials exhibit good linearity and large tuning range, which are preferable to phase change and flexible materials that highly rely on the nonlinear properties of nature materials.

Furthermore, metamaterials are also used in biosensing applications. The surface plasmonic metamaterial sensors pave the way to next-generation biological and chemical sensors for high-throughput, label-free, and multianalyte sensing applications [41,42]. The principle of surface plasmonic metamaterial for biosensing application is the resonant wavelength of absorption spectra shift by changing the medium refraction index. The metamaterial-based sensors are encapsulated and integrated with microfluidic chips to detect the biomolecules [43]. Such surface plasmonic metamaterial-based biosensors also suffer the same limitations of low sensitivity and selectivity [44].

In this study, we propose four designs of tunable split-disk metamaterial (SDM) absorbers. For a convenient description, SDM with one, two, three, and four splits are named SDM-1, SDM-2, SDM-3, and SDM-4, respectively. They are composed of a bottom Au mirror layer anchored on Si substrate and a suspended SDM nanostructure. These four SDMs are investigated first to determine the optimized geometrical parameters including split length and width. These four SDMs are reconfigurable and exhibit different active and flexible tunability in absorption intensity and resonance. Moreover, the resonances of the four SDMs can be tuned by changing the gap between the top SDM nanostructures and bottom Au mirror layers. When these SDMs are exposed to different environment ambients, they show high sensitivities with a high correlation coefficient of 0.9999. They are suitable for the use of gas sensors and biosensors.

## 2. Designs and Methods

The schematic drawing of proposed SDM is illustrated in Figure 1. It is composed of a bottom Au mirror layer anchored on Si substrate and a suspended SDM nanostructure. Four SDM configurations are periodic split-disks with one (SDM-1), two (SDM-2), three (SDM-3), and four (SDM-4) splits. The geometrical denotations are split length (*a_i_*) and split width (*b_i_*) as shown in Figure 1a, disk diameter (*d*), disk period (*P*), and the gap between bottom Au mirror layer and top SDM nanostructures (*z_i_*) as shown in Figure 1b, where *i* varies from 1 to 4 to represent SDM-1 to SDM-4, respectively. Herein, the *P*, *d* values, and thickness of Au layers are kept constant as 1.5 × 1.5 µm^2^, 1.0 µm, and 100 nm, respectively. The top SDM is suspended and supported by using a membrane composed of an Si/Si_3_N_4_ bilayer using residual stress induced electrothermal actuators (ETAs). As the structures are released, the initial stress between the Si and Si_3_N_4_ layers makes the cantilever deform out-of-plane and bend upward. By applying a direct current (DC) bias voltage on these residual stress-induced ETAs, the electrothermal attraction force is induced between the released cantilevers. Meanwhile, the out-of-plane bilayer cantilever will be bent downward to the substrate owing to such electrothermal attraction force. Therefore, the SDM can be tuned vertically by actuating the ETAs to change the *z* value between the top SDM and bottom mirror layer. When exposed on an electromagnetic wave, the absorber can generate the collective oscillation of free electrons on account of the metal-insulator-metal (MIM) structure [45,46], resulting in the absorption of incident electromagnetic energy.

To numerically calculate the optical properties of SDM devices, the Lumerical solution’s software finite difference time-domain (FDTD) solutions were employed in the full-field plane electromagnetic wave. Owing to the Si and Si_3_N_4_ bilayer with a very low absorption coefficient at the IR wavelength range, the influence of the electromagnetic wave propagation of the SDM can be ignored. The permittivity value of the Au material at the IR wavelength range is calculated according to the Drude-Lorentz model [47,48]. The propagation direction of the incident IR wave is set to be perpendicular to the x–y plane. The periodic boundary conditions are applied in the *x*- and *y*-axis direction, while the perfectly matched layer (PML) boundary conditions are assumed in the *z*-axis direction. The reflection (*R*) and the transmission (*T*) of the incident IR wave are calculated by spectra monitors set below and above the devices, respectively. The corresponding absorption (*A*) is defined as *A* = 1 − *T* − *R*. Since the transmission of incident electromagnetic light is blocked by the bottom Au mirror layer, the absorption can be simply obtained by *A* = 1 − *R*. It apparently means that high absorption is earned from low reflection.

## 3. Results and Discussion

Figure 2 shows the absorption spectra of four SDMs with different *a* values, where *z* and *b* values are kept constant as *z* = 100 nm, and *b* = 200 nm, respectively. SDM-1 and SDM-3 exhibit dual-resonance characteristics, which are denoted as *ω*_1_ and *ω*_2_ as shown in Figure 2a,c, while SDM-2 and SDM-4 show only single-resonance as shown in Figure 2b,d, respectively. It is obvious to observe that *ω*_2_ is the main resonance for SDM-1 and SDM-3 since they exhibit the identical tuning characteristic of SDM-2 and SDM-4. By changing *a* values from 100 nm to 500 nm for SDM-1 and SDM-2 and from 100 nm to 400 nm for SDM-3 and SDM-4, the main absorption resonances are red-shifted from the wavelengths of 2.76 μm to 4.09 μm, 2.78 μm to 4.38 μm, 2.65 μm to 3.32 μm, and 2.70 μm to 3.56 μm for SDM-1, SDM-2, SDM-3, and SDM-4, respectively. While the absorption intensities increase to 99.8%, 75.5%, 52.5%, and 47.8% for SDM-1, SDM-2, SDM-3, and SDM-4, respectively. It is worth noting that the main resonances disappear when *a* = 500 nm for SDM-2 and *a* = 400 nm for SDM-3 and SDM-4 because of the enlarged splits contacting each other and then separating the disks.

Figure 3 shows the absorption spectra of four SDMs with different *b* values, where *z* and *a* values are kept constant as *z* = 100 nm, and *a* = 300 nm. The absorption resonances of *ω*_1_ are blue-shifted by increasing *b* values for SDM-1 and SDM-3 as shown in Figure 3a,c, respectively. The tuning ranges are 230 nm from the wavelength of 1.80 μm to 2.03 μm for SDM-1 and from the wavelength of 2.23 μm to 2.46 μm for SDM-3, while the main absorption resonances of the four SDMs are red-shifted by increasing the *b* values from 100 nm to 300 nm and then blue-shifted by increasing the *b* values from 300 nm to 500 nm. The tuning ranges of the main absorption resonances are 80 nm, 210 nm, 160 nm, and 150 nm for SDM-1, SDM-2, SDM-3, and SDM-4, respectively. The main resonances disappear when *b* = 400 nm for SDM-3 and SDM-4 since the disks are separated. Therefore, the shifts of the main absorption resonances are dominated by *a* values in the ranges of 1330 nm, 1600 nm, 670 nm, and 860 nm for SDM-1, SDM-2, SDM-3, and SDM-4, respectively, and gradually enlarge the absorption intensities, while the shifts of the main absorption resonances are minor by changing *b* values.

To test the SDMs’ possession of active tunability, the top SDM nanostructures are suspended and connected to the MEMS actuators for the changing *z* values of the SDMs. Figure 4 shows the absorption spectra of four SDMs with different *z* values, where *a* and *b* values are kept constant as *a* = 300 nm, and *b* = 200 nm. The gaps could be actively tuned by using electrostatic or electrothermal forces. The maximum absorption intensities of the main resonances are 99.9%, 82.8%, 91.5%, and 85.8% at the wavelengths of 3.42 μm, 3.82 μm, 3.56 μm, and 3.80 μm for SDM-1 (Figure 4a), SDM-2 (Figure 4b), SDM-3 (Figure 4c), and SDM-4 (Figure 4d), respectively. The *z* values are 50 nm. Under these conditions, SDM-1 exhibits perfect absorption up to 99.9%, indicating its potential application as a high-efficient absorber. By changing the *z* value from 50 nm to 200 nm, the absorption resonances of the SDMs are blue-shifted in the range of 320 nm. In addition, the absorption intensities could be attenuated and the spectra bandwidths become broader. This indicates the proposed SDMs could be explored for variable optical attenuator (VOA) application.

To further prove the proposed SDMs for sensing applications, the four SDMs were designed to be exposed to ambient environments with different refraction indexes (*n*_*i*_) to demonstrate the high-efficient and high-sensitive sensor applications. Figure 5 shows the absorption spectra of the SDMs by changing the surrounding refraction index from 1.0 to 1.4, where the *z*, *a*, and *b* values are kept constant as *z* = 50 nm, *a* = 300 nm, and *b* = 200 nm, respectively. SDM-1 shows the absorption resonances are red-shifted from the wavelength of 2.04 μm to 2.83 μm and from the wavelength of 3.42 μm to 4.74 μm for *ω*_1_ and *ω*_2_, respectively, by increasing the *n*_1_ value from 1.0 to 1.4 as shown in Figure 5a. The absorption intensities are stable at 99.9%. As shown in Figure 5b, the absorption resonances are red-shifted from the wavelength of 3.81 μm to 5.29 μm for SDM-2 by increasing the *n*_2_ value, while the absorption intensities increase minorly from 82.8% to 89.5%. SDM-3 shows the absorption resonances are red-shifted from the wavelength of 2.54 μm to 3.46 μm and from the wavelength of 3.56 μm to 4.89 μm for *ω*_1_ and *ω*_2_, respectively, by increasing the *n*_3_ value as shown in Figure 5c. The absorption intensities increase gradually from 64.9% to 75.3% and from 91.4% to 95.4% for *ω*_1_ and *ω*_2_, respectively. As for SDM-4, the absorption resonances are red-shifted from the wavelength of 3.80 μm to 5.24 μm by increasing the *n*_4_ value, while absorption intensities increase minorly from 85.8% to 91.1% as shown in Figure 5d. The four SDMs exhibit refraction index-sensitive features that indicate that absorption resonances are red-shifted by increasing *n_i_* values. It can be explained that the relative permittivity of the surrounding insulator layer changed, resulting in the change of resonant frequency [17].

To further evaluate the sensitivity (*S*) and figure of merit (FOM) of SDM-based sensors, they can be defined as follows [49].
(1)$$S=\frac{\Delta \lambda }{\Delta n}\;\;,\;\;\;FOM=\frac{S}{FWHM}$$
where Δ*λ* is the resonance shift caused by a refraction index change (Δ*n*) and FWHM is the full width at half maximum (FWHM) of resonance. Figure 6 shows the relationships of resonances and *n*_*i*_ values of the SDMs. The fitting trends are highly linear, with a correlation coefficient of 0.9999. The *S* and FOM values are calculated at main resonances. The four SDMs exhibit high sensitivity of 3312 nm/RIU, 3362 nm/RIU, 3342 nm/RIU, and 3567 nm/RIU as shown in Figure 6a–d, respectively. These results are better than those reported in this wavelength range [50,51,52,53], where the maximum sensitivity is only 1060 nm/RIU. The maximum FOM values are 13.25, 15.28, 20.89, and 17.8 for SDM-1, SDM-2, SDM-3, and SDM-4, respectively. These results indicate the proposed SDM absorbers are suitable to be used as high-efficient and high-sensitive gas sensors and biosensors, since the refraction index of the ambient environment could be calculated by the change in the absorption resonances, which is related to the type of gas and bio-molecules.

To better understand the coupling effects within SDMs, the corresponding electric (*E*) field and magnetic (*H*) field distributions of SDMs are plotted in Figure 7 and Figure 8, respectively, where the *n*, *z*, *a*, and *b* values are kept constant as *n* = 1.0, *z* = 50 nm, *a* = 300 nm, and *b* = 200 nm, respectively. In the main resonances, E-field energies are focused on the edges of disks and insulator layers as shown in Figure 7b,c,e,f while H-field energies are confined in insulator layers right below the center of the disks as shown in Figure 8b,c,e,f. The electromagnetic energies are confined within the top SDM nanostructures and bottom Au mirror layers, and the observed field energy around the edge of the SDMs are electric dipole resonances. The electrical vectors of the up and down sides of the top of the SDM, and left and right parts of the insulator layers are opposite, respectively, resulting in a strong magnetic response. The combination of strong local magnetic response and electric dipole resonances cause the enhanced absorption. The physical mechanism could be explained by an LC oscillator circuit excited by an external electromagnetic field. The equivalent circuit model of MIM nanostructures are inductances, resistances, and capacitors, which is equivalent to an LC oscillator circuit. According to the Drude-Lorentz model [17], the resonant frequency can be expressed by
(2)$$f_{LC}=\frac{1}{2\pi \sqrt{L_{e}C_{e}}}=\left(\frac{C_{0}}{2\pi l\sqrt{\varepsilon_{c}}}\right)\sqrt{\frac{z}{w}}\;$$
where *L_e_* = *μ*0*l*2/*t* and *C_e_* = *ε*0*ε**_c_wt/z* are the equivalent inductance and capacitance determined by SDM cells, *C*_0_ is the light velocity in vacuum, *ε_c_* is the relative permittivity of the materials within the SDM, *w* is the metallic width, *z* is the gap width, *l* is the size of the circuit model (i.e., *a* and *b*), and *t* is the metallic thickness. For *ω*_1_, the E- and H-field energies are focused nearby or below the edge of the top of the SDM for SDM-1 as shown in Figure 7a and Figure 8a, respectively, while the E- and H-field energies are confined near the splits for SDM-3 as shown in Figure 7d and Figure 8d, respectively. This demonstrates that absorption resonances *ω*_1_ are caused by the coupling effects within the top of the SDM nanostructures.

## 4. Conclusions

In conclusion: we propose four tunable SDM absorbers. By tailoring the geometrical dimensions, the absorption spectra could be tuned within a “molecule fingerprint” wavelength range, which indicates the potential application as multi-gases sensors and biosensors. Furthermore, the absorption intensities decrease by increasing the gap between the SDMs and bottom Au mirror layer. To demonstrate that the proposed SDM devices could be applied as high-sensitive sensors, the SDM devices were exposed to ambient environments with different refraction indexes. The four SDMs exhibit high sensitivity of 3312 nm/RIU, 3362 nm/RIU, 3342 nm/RIU, and 3567 nm/RIU for SDM-1, SDM-2, SDM-3, and SDM-4, respectively. The corresponding FOM are 13.25, 15.28, 20.89, and 17.8 for SDM-1, SDM-2, SDM-3, and SDM-4, respectively. The correlation coefficient is 0.9999. The tuning range of resonance is within the range of a “molecule fingerprint”, allowing the SDM devices to be used for the identification of molecules. These results open an avenue to their application as high-efficient multi-gas sensors and biosensors, switches, VOAs, and so on.

## Figures and Tables

**Figure 1 nanomaterials-11-00598-f001:**
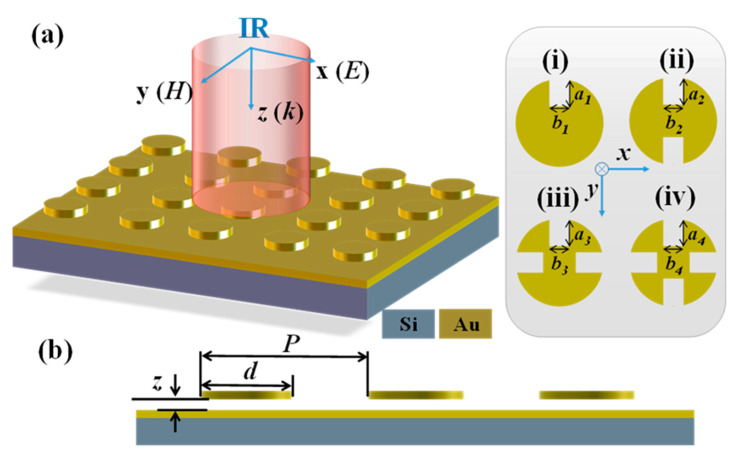
(**a**) Schematic drawing of four types of split-disk metamaterials (SDMs). They are (i) SDM-1, (ii) SDM-2, (iii) SDM-3 and (iv) SDM-4, respectively. (**b**) The cross-sectional view of SDM.

**Figure 2 nanomaterials-11-00598-f002:**
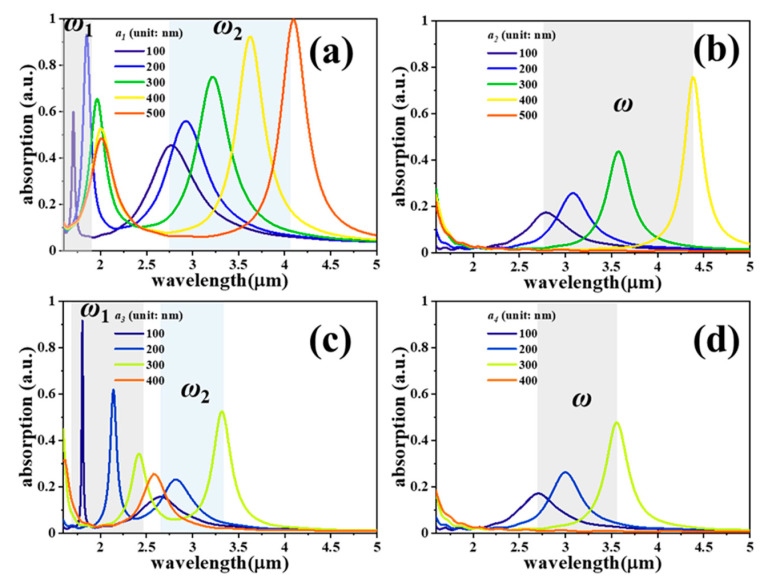
Absorption spectra of (**a**) SDM-1, (**b**) SDM-2, (**c**) SDM-3, and (**d**) SDM-4 with different *a* values, where *z* and *b* values are kept constant as *z* = 100 nm, and *b* = 200 nm, respectively.

**Figure 3 nanomaterials-11-00598-f003:**
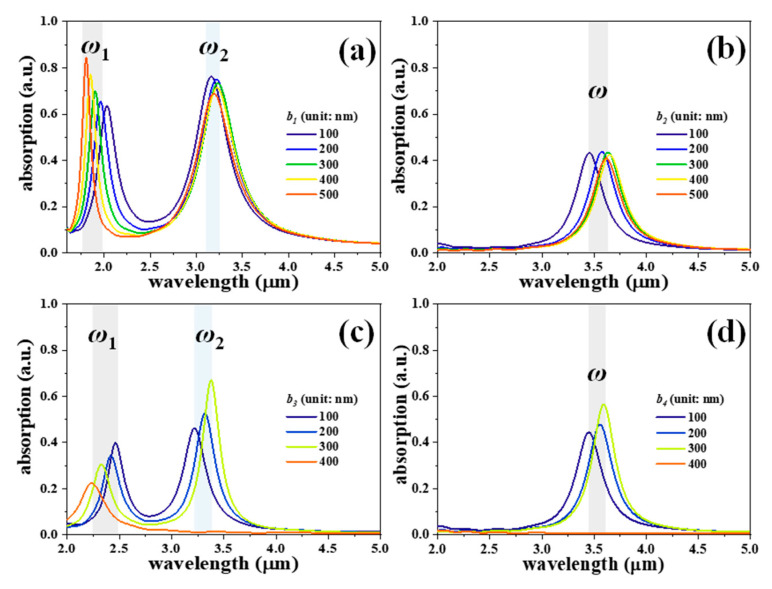
Absorption spectra of (**a**) SDM-1, (**b**) SDM-2, (**c**) SDM-3, and (**d**) SDM-4 with different *b* values, where *z* and *a* values are kept constant as *z* =100 nm, and *a* = 300 nm, respectively.

**Figure 4 nanomaterials-11-00598-f004:**
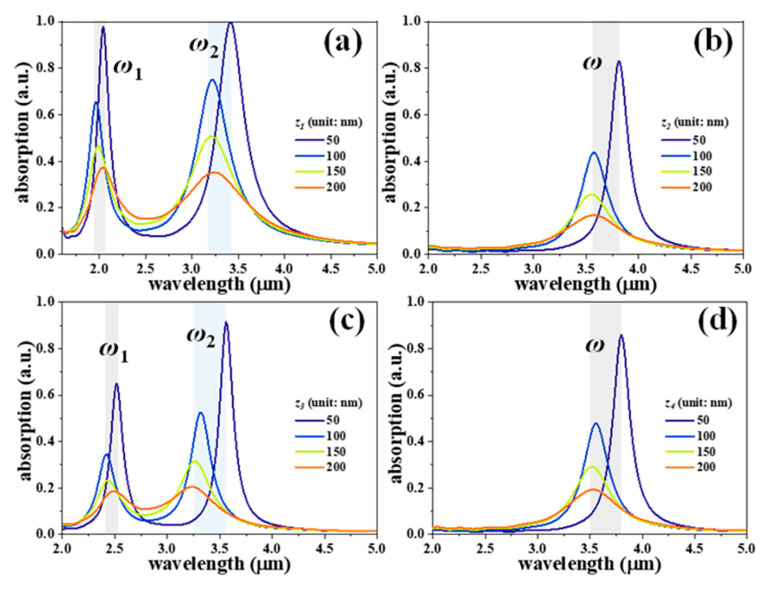
Absorption spectra of (**a**) SDM-1, (**b**) SDM-2, (**c**) SDM-3, and (**d**) SDM-4 with different *z* values, where *a* and *b* values are kept constant as *a* = 300 nm, and *b* = 200 nm, respectively.

**Figure 5 nanomaterials-11-00598-f005:**
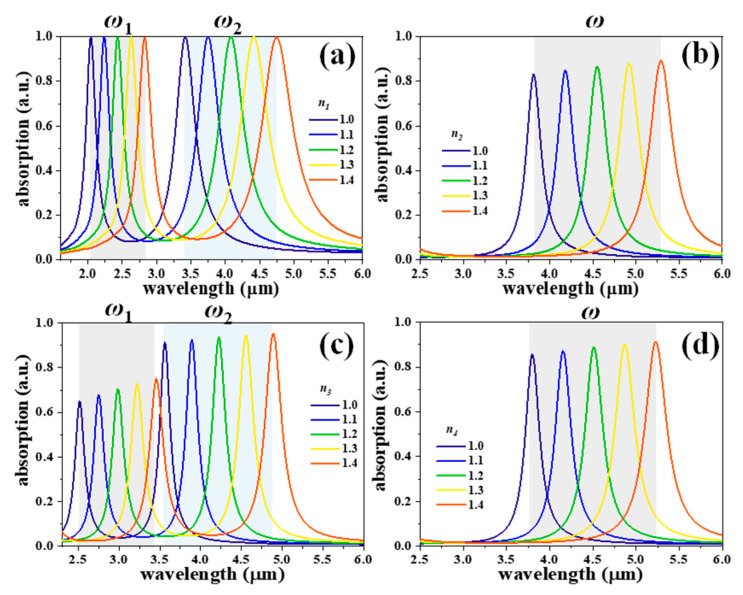
Absorption spectra of (**a**) SDM-1, (**b**) SDM-2, (**c**) SDM-3, and (**d**) SDM-4 exposed to ambient environments with different refraction indexes, where *z*, *a*, and *b* values are kept constant as *z* = 50 nm, *a* = 300 nm, and *b* = 200 nm, respectively.

**Figure 6 nanomaterials-11-00598-f006:**
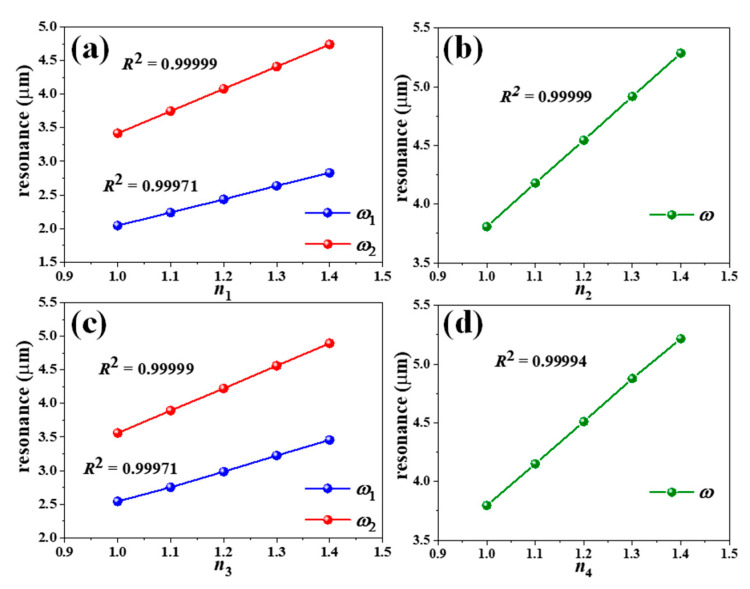
The relationships of resonances and *n* values of (**a**) SDM-1, (**b**) SDM-2, (**c**) SDM-3, and (**d**) SDM-4, respectively.

**Figure 7 nanomaterials-11-00598-f007:**
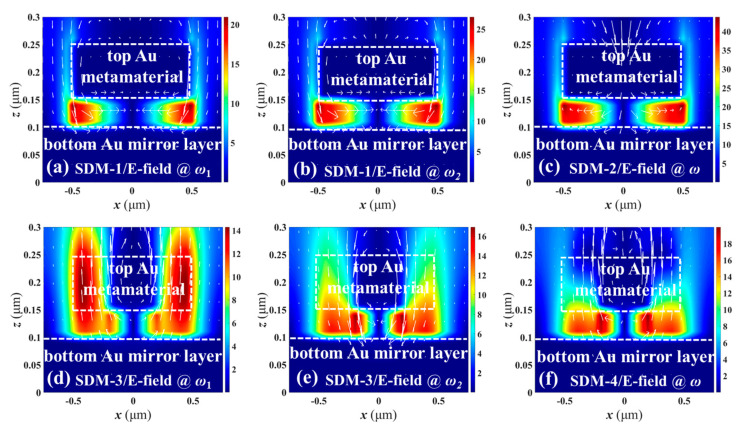
E-field distributions of SDM-1 monitored at (**a**) ω_1_ and (**b**) ω_2_ resonances, SDM-2 monitored at (**c**) ω resonance, SDM-3 monitored at (**d**) ω_1_ and (**e**) ω_2_ resonances, and SDM-4 monitored at (**f**) ω resonance, respectively.

**Figure 8 nanomaterials-11-00598-f008:**
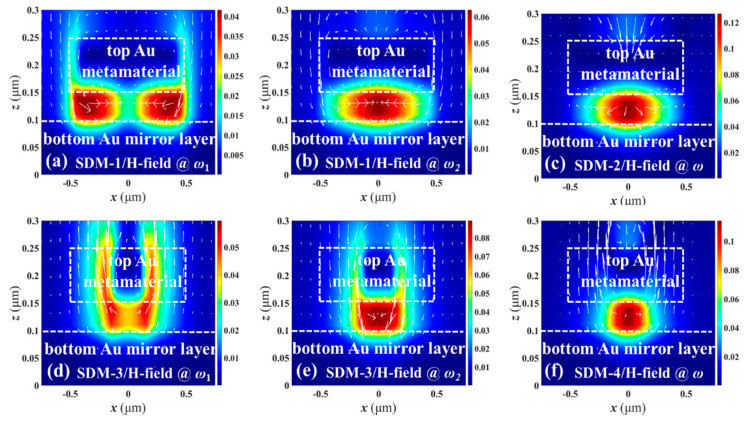
H-field distributions of SDM-1 monitored at (**a**) ω_1_ and (**b**) ω_2_ resonances, SDM-2 monitored at (**c**) ω resonance, SDM-3 monitored at (**d**) ω_1_ and (**e**) ω_2_ resonances, and SDM-4 monitored at (**f**) ω resonance, respectively.

## Data Availability

Not applicable.

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
