# Peer review of "Tunable Split-Disk Metamaterial Absorber for Sensing Application"

_nanomaterials, 2021, doi:10.3390/nano11030598_

Round 1

Reviewer 1 Report

In the current work, the authors fabricate and study four types of split-disk metamaterial absorbers. By changing the geometry, the nanostructures exhibit different absorption resonances in the range from 1.5 to 5.0 μm wavelengths. The manuscript is clearly written; the conclusions are in general supported by the data shown, although additional measurements are required. The current work contributes to the field that is interesting for many readers of the journal. However, several issues listed below should be amended prior to publication.

The key major concern that I have is related to the novelty of the current work. Whereas the authors present a new geometry of plasmonic nanostructures that to my knowledge has not been shown yet, I have not found any argument why is it better than many other nanostructures that have been studied before. Although the authors site many of the relevant papers, there are many more recent articles where similar nanostructures where studied. In my view, the authors should clearly answer the question why do they think this geometry is better/easier to produce, etc.

Second, I believe that a proper characterization of new types of metamaterials is hardly possible without their direct visualization using SEM imaging. The authors could show a few examples of images of the nanostructures studied.

Minor points:

RIU should be spelled out

Figures 2-5: absorbance is a logarithm and hence, is a dimensionless value; therefore no a.u. is needed for the y-axes.

Author Response

Please refer to the attached response letter and revised manuscript. 

Reviewer 2 Report

Numerical design of split-disk metamaterial (SDM) absorbers is presented.

The topic is appropriate for publication in the journal. The question is original but not well defined. First, it is not clear in the presentation that the paper deals with a numerical study.

Then, the scientific background is not well presented: it is not clear if the proposed metamaterials have already been studied in the literature (theoretically or experimentally) or they are originally proposed by the authors. It is not clear the experimental feasibility of the theoretical design. In section 2 and 3 authors declare: “The gaps could be actively tuned by using electrostatic or electrothermal forces”. It is not clear the feasibility of the integration of such technologies with realization of the proposed metamaterial. Is there some reference with experimental results supporting this guess?

In section 3 the authors declare: “SDM-1 exhibits perfect absorption”. I think this observation deserve a specific comment: it seems an important aspect to be underlined and discussed better.

Always in section 3, the authors say that the refractive index range of investigation is between 1 and 1.4. It is a wide range not really specific for gas sensing or biosensing. Comparison of sensitivity parameters with a different optical sensor working in this range for the specific application could be useful to understand the sensing performance of the proposed transducer.

Figures 7 and 8 attest that “The electromagnetic energies are confined within the top SDM nanostructures and bottom Au mirror layers”. This aspect is not geometrically favourable for gas or biosensing purposes owing to the experimental difficulty in immobilizing the sensing surface with the specific receptor. Have the authors taken in consideration this drawback?

Finally the conclusion that the proposed metamaterial “tuned spanned the perfect zone of “molecule fingerprint” should be scaled back as there is not enough experimental evidences of this statement.

Minor comments: English should be improved in lines: 125, 152-153, 204-205.

Author Response

(The authors gave the same response as above.)

Round 2

Reviewer 1 Report

The authors have provided reasonable answers to my questions. In my view, at least a brief version of the discussion that is provided in the response letter can be included to the manuscript (which is not done yet). Once it is done, I have no further critics.

Author Response

Dear Sir/Madam,

Thank you very much for accepting our revised manuscript and your supervision of the reviewing process of this manuscript. We have revised as red portions in the revised manuscript. 

Best Regards,

Dr. Yu-Sheng Lin

Reviewer 2 Report

Revision of the paper has satisfied my doubts.

As for me, it can be accepted for publication.

Author Response

Dear Sir/Madam,

Thanks for your acception. 

Best Regards,

Dr. Yu-Sheng Lin